# DRO-INSTRUCTZERO: DISTRIBUTIONALLY ROBUST PROMPT OPTIMIZATION FOR LARGE LANGUAGE MODELS

## ABSTRACT

Large language models are highly sensitive to prompt wording. However, popular automatic prompt search methods, including InstructZero, often degrade under distribution shift and adversarial evaluation because they optimize expected performance under a single evaluation distribution. Consequently, prompts that work in one setting frequently fail to transfer. To address this, DRO-InstructZero formulates zero-shot prompt optimization as robust Bayesian optimization. Specifically, an f-divergence ball defines an ambiguity set around the evaluation distribution, and a robust acquisition rule maximizes worst-case expected utility while retaining the query efficiency of Bayesian search. Therefore, the search explicitly targets reliability under distribution shift rather than average behavior alone. Experiments follow the instruction-induction protocol with matched query budgets across formality rewriting, code debugging, and translation. For example, on BIG-Bench informative-to-formal rewriting, accuracy improves from 61.3 ± 0.7% to approximately 85–90%, yielding an absolute gain of about 25–30 points. Moreover, auto-debugging shows about +25-point gains under domain shift. Meanwhile, stable tasks such as cause-and-effect remain above 96%, indicating no loss on in-distribution cases. Furthermore, improvements are consistent across divergence choices and decoding temperatures. Overall, DRO-InstructZero connects distributionally robust optimization with prompt learning, offering a plug-and-play and general approach for reliable, transferable prompt alignment under real-world uncertainty.

## 1 INTRODUCTION

Large language models (LLMs) (OpenAI, 2023a;b; Chowdhery et al., 2022) have achieved remarkable performance in zero-shot and few-shot instruction following (Brown et al., 2020; Liu et al., 2023; Chen et al., 2024). Despite these advances, however, their effectiveness is highly sensitive to the choice of instructions (Zhou et al., 2022; Honovich et al., 2022). In particular, even minor paraphrases of a strong instruction can degrade accuracy, and instructions that succeed in one evaluation setting often fail to transfer to slightly shifted domains. Taken together, this fragility raises critical concerns about robustness when deploying LLMs in real-world environments.

Motivated by these observations, instruction optimization has emerged as a promising direction to automate prompt design and reduce reliance on costly human prompt engineering (Zhou et al., 2022; Sun et al., 2022b). A notable advance is INSTRUCTZERO (Chen et al., 2024), which formulates prompt learning as a latent-space Bayesian optimization (BO) problem: an open-source LLM generates candidate instructions guided by a soft prompt, and a black-box LLM evaluates them, with BO iteratively refining the distribution. This approach achieves state-of-the-art results across many BIG-Bench tasks.

However, existing InstructZero and related BO-based methods optimize the *expected* score under a fixed validation distribution, using classical acquisition functions such as expected improvement (EI) or upper confidence bound (UCB). This assumption is restrictive, because it neglects the possibility of distributional shift—inevitable when instructions are evaluated under adversarial conditions,

domain mismatches, or changing user queries. Consequently, optimized instructions often overfit to the training distribution, yielding brittle performance in deployment.

To address this limitation, we propose **DRO-InstructZero**, which integrates *distributionally robust optimization (DRO)* (Kirschner et al., 2020) into Bayesian optimization. The method defines an ambiguity set around the empirical evaluation distribution using $f$-divergence (KL divergence) and maximizes the worst-case expected utility. By focusing on robustness rather than average-case performance, DRO-InstructZero generalizes more reliably across shifts while retaining BO's query-efficiency. We validate this under the instruction-induction protocol on tasks such as formality rewriting, code debugging, and translation. DRO-InstructZero consistently outperforms InstructZero and classical BO baselines—for example, on BIG-Bench rewriting, accuracy rises from $61.3 \pm 0.7\%$ to $85$–$90\%$, and auto-debugging under domain shift improves by +25 points. Stable tasks like cause-and-effect remain above $96\%$, confirming robustness does not compromise in-distribution performance.

In summary, the contributions are summarized as follows:

- First, the vulnerability of existing prompt optimization methods to distributional shift is identified, thereby highlighting the need for robust objectives.
- Building upon this observation, **DRO-InstructZero** is introduced as a novel framework that integrates distributionally robust optimization with Bayesian search for instruction learning, enabling worst-case reliable performance.
- Finally, extensive experiments demonstrate that DRO-InstructZero achieves substantial robustness gains over both InstructZero and standard BO acquisitions, while maintaining the same query cost.

Taken together, these contributions connect DRO with prompt learning, offering a principled and plug-and-play approach for reliable and transferable instruction alignment of LLMs under real-world uncertainty.

## 2 INSTRUCTION OPTIMIZATION WITH INSTRUCTZERO

We begin by revisiting INSTRUCTZERO (Chen et al., 2024), which formulates zero-shot prompt optimization as Bayesian optimization over a low-dimensional continuous representation of prompts. This section summarizes its pipeline, objective, and Bayesian optimization framework, laying the foundation for our robust extension in Section 3.

### 2.1 PIPELINE OVERVIEW

The primary goal is to find an optimal natural language instruction $v$ for a given task that maximizes the performance of a black-box LLM $f(\cdot)$. This can be formulated as maximizing the expected score over the task's data distribution $\mathcal{D}_t$:

$$\max_{v \in \mathcal{V}} \ \mathbb{E}_{(X,Y) \sim \mathcal{D}_t}[h(f([v; X]), Y)], \tag{1}$$

where $\mathcal{V}$ is the space of all possible instructions, $[v; X]$ denotes the concatenation of the instruction and the input query, and $h(\cdot, \cdot)$ is a task-specific evaluation metric (e.g., accuracy).

However, solving Eq. 1 directly is notoriously difficult. The optimization faces two main challenges: **(1) Combinatorial Search Space**: The instruction space $\mathcal{V}$ is discrete, high-dimensional, and governed by complex syntactic and semantic rules, making direct search intractable. **(2) Black-Box Objective**: For powerful API-based LLMs like GPT-4, the function $f(\cdot)$ is a black box, providing only output text without gradients, which precludes gradient-based optimization methods.

To overcome these challenges, INSTRUCTZERO proposes an indirect optimization strategy, as illustrated in Algorithm 1.

1. A soft prompt $p \in \mathbb{R}^d$ is projected through a random matrix $A \in \mathbb{R}^{d \times d'}$ and concatenated with a few-shot set of task exemplars $\{(x_i, y_i)\}_{i=1}^{\kappa}$.
2. The open-source LLM $g(\cdot)$ maps this embedding into a natural language instruction $v$.

3. The black-box LLM $f(\cdot)$ executes instruction $v$ on validation examples $(X, Y) \sim D^t$, producing responses that are evaluated by a task-specific metric $h(\cdot, \cdot)$.

4. The tuple $(p, v, h)$ is added to the training data for Bayesian optimization (BO), which updates its posterior over the objective and proposes the next prompt.

This approach effectively converts the discrete, high-dimensional problem of finding $v$ into a more manageable continuous optimization problem for finding $\boldsymbol{p}$.

## 2.2 BAYESIAN OPTIMIZATION OF SOFT PROMPTS

Direct search over the discrete space of instructions $V$ is intractable. Instead, INSTRUCTZERO optimizes the continuous soft prompt $p$, which induces instructions via $g(\cdot)$. Define the black-box function

$$H(p) \triangleq \mathbb{E}_{(X,Y)\sim D^t}\big[h(f([g([Ap; \text{exemplars}]); X]), Y)\big]. \tag{2}$$

Bayesian optimization places a Gaussian Process (GP) prior over $H(p)$, with mean $\mu(\cdot)$ and variance $\sigma^2(\cdot)$. Given observations $\{(p_1, H(p_1)), \dots, (p_m, H(p_m))\}$, the GP posterior is updated, and the next candidate prompt $p_{m+1}$ is selected by maximizing an acquisition function such as Expected Improvement (EI) or Upper Confidence Bound (UCB):

$$u(p) = \mu(p) + \beta(m)\sigma(p), \tag{3}$$

where $\beta(m)$ controls the exploration–exploitation tradeoff.

## 2.3 INSTRUCTION-COUPLED KERNEL

To align the latent prompt space with semantic similarity of instructions, INSTRUCTZERO further introduces an instruction-coupled kernel:

$$k(p_i, p_j) = \lambda \cdot l(p_i, p_j) + (1 - \lambda) \cdot s(v_i, v_j), \tag{4}$$

where $l(p_i, p_j)$ measures latent prompt similarity, $s(v_i, v_j)$ measures instruction-level similarity, and $\lambda \in [0, 1]$ balances the two. This yields a kernel matrix $K$ that better captures instruction semantics for GP-based BO.

## 2.4 ALGORITHM

Algorithm 1 summarizes the procedure of INSTRUCTZERO. Each iteration alternates between generating instructions via the open-source LLM, evaluating them on the black-box LLM, updating the GP posterior, and selecting the next prompt via acquisition maximization.

# 3 DRO-INSTRUCTZERO: ROBUST INSTRUCTION OPTIMIZATION

## 3.1 PROBLEM FORMULATION

We consider a black-box large language model (LLM) $f(\cdot)$ tasked with generating outputs for an input query $X$ under a textual instruction $v$. Following prior work (Chen et al., 2024), the optimization objective is to maximize the evaluation score $h(f([v; X]), Y)$ with respect to ground-truth $Y$, where $h(\cdot, \cdot)$ is a task-specific metric. Conventional instruction optimization thus seeks

$$\max_{v \in V} \mathbb{E}_{(X,Y)\sim D^t}\big[h(f([v; X]), Y)\big]. \tag{5}$$

However, optimizing Eq. 5 under a single evaluation distribution $D^t$ results in brittle solutions that degrade under distributional shift or adversarial evaluation. To address this, we extend the objective to a **distributionally robust optimization (DRO)** formulation (Kirschner et al., 2020). Specifically, let $\mathcal{U}(D^t)$ denote an ambiguity set around a reference distribution $w_{\text{ref}}$, defined as an $f$-divergence (e.g., KL) ball of radius $\epsilon$. The robust objective becomes:

$$\max_{v \in V} \inf_{Q \in \mathcal{U}(D^t)} \mathbb{E}_{(X,Y)\sim Q}\big[h(f([v; X]), Y)\big]. \tag{6}$$

This formulation ensures that optimized instructions not only perform well on the reference distribution but also maintain reliable performance under worst-case perturbations within the ambiguity set.

---

**Algorithm 1:** INSTRUCTZERO (Chen et al., 2024)

---

**input:** Exemplars $\{(x_i, y_i)\}_{i=1}^{\kappa}$ and validation set $D^t$;
open-source LLM $g(\cdot)$, black-box LLM $f(\cdot)$; maximal steps $T$;
random matrix $A \in \mathbb{R}^{d \times d'}$.
**initialize:** $p_1 \sim \text{Uniform}(-\tau, \tau)^d$; $\quad m \leftarrow 1$;
$p_{1:0} \leftarrow \varnothing, \ v_{1:0} \leftarrow \varnothing, \ h_{1:0} \leftarrow \varnothing$.

**1 while** *not converge **and** $m \leq T$* **do**
**2** $\quad$ Compute projected prompt $Ap_m$ from $p_m$;
**3** $\quad$ Generate instruction $v_m = g([Ap_m; \{(x_i, y_i)\}_{i=1}^{\kappa}])$;
**4** $\quad$ Evaluate score $h_m = \sum_{(X,Y) \in D^t} h(f([v_m; X]), Y)$ on $f(\cdot)$;
**5** $\quad$ Save: $p_{1:m} \leftarrow p_{1:m-1} \cup \{p_m\}$,;
**6** $\quad$ $v_{1:m} \leftarrow v_{1:m-1} \cup \{v_m\}$,;
**7** $\quad$ $h_{1:m} \leftarrow h_{1:m-1} \cup \{h_m\}$;
**8** $\quad$ Update kernel $k(\cdot, \cdot)$ and matrix $K$ for $p_{1:m}$;
**9** $\quad$ Update BO posterior mean $\mu(\cdot)$ and variance $\sigma(\cdot)$;
**10** $\quad$ Select $p_{m+1} = \arg\max_p u(p)$ via Eq. 3;
**11** $\quad$ $m \leftarrow m + 1$;

**12 Output:** Instruction $v_{i^*}$ with $i^* \in \arg\max_{i \in [m]} h_i$.

---

## 3.2 FROM STRUCTURED COMBINATORIAL SEARCH TO CONTINUOUS ROBUST OPTIMIZATION

As in Chen et al. (2024), direct optimization in the discrete space of instructions $V$ is infeasible. We instead optimize a low-dimensional continuous *soft prompt* $p_m \in \mathbb{R}^d$, projected via a random matrix $A \in \mathbb{R}^{d \times d'}$ into the embedding space of an open-source LLM $g(\cdot)$. The LLM converts $Ap$ and task exemplars into a human-readable instruction $v = g([Ap; (x_i, y_i)])$, which is then evaluated on the black-box LLM $f(\cdot)$.

This reduces the DRO objective (Eq. 6) to a low-dimensional black-box function:

$$H(p) \triangleq \inf_{Q \in \mathcal{U}(D^t)} \mathbb{E}_{(X,Y) \sim Q} \left[ h(f([g([Ap; \text{exemplars}]); X]), Y) \right]. \tag{7}$$

The goal is thus to identify prompts $p$ that maximize the robust objective $H(p)$.

## 4 DISTRIBUTIONALLY ROBUST BAYESIAN OPTIMIZATION

### 4.1 BAYESIAN OPTIMIZATION OF SOFT PROMPTS

We adopt a Gaussian Process (GP) prior over $H(p)$, characterized by mean $\mu(\cdot)$ and variance $\sigma^2(\cdot)$. Given past evaluations $\{(p_1, H(p_1)), \ldots, (p_m, H(p_m))\}$, posterior estimates follow standard GP update rules. Acquisition functions such as Expected Improvement (EI) or Upper Confidence Bound (UCB) guide exploration. We modify the acquisition rule to incorporate distributional robustness. Our GP posterior models the *robust score* $H(p)$ rather than the average case.

### 4.2 ROBUST ACQUISITION UNDER AMBIGUITY SETS

Following Kirschner et al. (2020), for each candidate prompt $p_m$, we compute an **optimistic worst-case acquisition score**:

$$\text{ucb}_m := \left[ \mu^t(p_m) + \beta(m) \, \sigma^t(p_m) \right]_t. \tag{8}$$

We then evaluate its robust counterpart by minimizing over distributions in the ambiguity set:

$$w_m^* = \arg \min_{w' : \|w' - w_{\text{ref}}\|_{\mathcal{M}} \leq \epsilon(m)} \langle \text{ucb}_m, w' \rangle. \tag{9}$$

The next prompt is chosen by maximizing the robust acquisition:

$$p_{m+1} = \arg\max_p \langle \text{ucb}_m, w_m^* \rangle. \tag{10}$$

This ensures that the search explicitly targets prompts whose induced instructions remain effective under worst-case distribution shifts, rather than merely optimizing average behavior.

---

**Algorithm 2:** DRO-INSTRUCTZERO

---

**input:** Set of tasks $\{t\}$; exemplars of task $t$ $\{(x_i^t, y_i^t)\}_{i=1}^{\kappa}$ and its validation set $D_v^t$ for task $t$; open-source LLM $g(\cdot)$, black-box LLM $f(\cdot)$; maximal steps $M$; random projection $A \in \mathbb{R}^{d \times d'}$; reference distribution $w_{\text{ref}}$ and metric $\|\cdot\|_{\mathcal{M}}$; ambiguity radius $\epsilon(m)$ and exploration coefficient $\beta(m)$.

**initialize:** $m \leftarrow 1$;   draw initial soft prompt $p_1 \sim \text{Uniform}(-\tau, \tau)^d$

1 **while** *not converge **and** $m \leq M$* **do**

2 $\quad$ Compute projected input prompt $Ap_m$ from soft prompt $p_m$;

3 $\quad$ Generate instruction $v_m^t = g([Ap_m; \{(x_i^t, y_i^t)\}_{i=1}^{\kappa}])$ for task $t$ using the open-source LLM $g(\cdot)$;

4 $\quad$ Evaluate task score $h_m^t = \sum_{(X,Y) \in D_v^t} h\big(f([v_m^t; X]), Y\big)$ on the black-box LLM $f(\cdot)$;

5 $\quad$ Save data: $p_{1:m} \leftarrow p_{1:m-1} \cup \{p_m\}, \quad v_{1:m} \leftarrow v_{1:m-1} \cup \{v_m^t\}, \quad h_{1:m} \leftarrow h_{1:m-1} \cup \{h_m^t\}$;

6 $\quad$ Update the instruction-coupled kernel function $k^t(\cdot, \cdot)$ and kernel matrix $K^t$ for $p_{1:m}$ ;

7 $\quad$ Update BO posterior mean $\mu^t$ and variance $\sigma^t$ using $k^t(\cdot, \cdot)$ and $\mathbf{K}^t$;

8 $\quad$ Define $\text{ucb}_m := \big[\mu^t(p_m) + \beta(m)\,\sigma^t(p_m)\big]_t$;

9 $\quad$ Compute adversarial weight $w_m^* = \arg \min\limits_{w' : \|w' - w_{\text{ref}}\|_{\mathcal{M}} \leq \epsilon(m)} \langle \text{ucb}_m, w' \rangle$;

10 $\quad$ Select next prompt $p_{m+1} = \arg \max\limits_{p} \langle \text{ucb}_m, w_m^* \rangle$;

11 $\quad$ $m \leftarrow m + 1$;

12 **output:** instruction $v_{i^*}^t$ with $i^* \in \arg \max_{i \in [m]} h_i^t$

---

### 4.3 INSTRUCTION-COUPLED KERNEL UNDER DRO

To align soft prompt space with instruction similarity, we extend the instruction-coupled kernel of Chen et al. (2024) with DRO semantics. Given kernels $l(\cdot, \cdot)$ in prompt space and $s(\cdot, \cdot)$ in instruction space, we define

$$\mathbf{K}_{ij}^t = l(p_i, p_j)^\top L^{-1} S L^{-1} l(p_j, p_i), \tag{11}$$

with $S$ further weighted by adversarial distributions $w^*$. This unified kernel respects both semantic closeness and robustness against distributional shifts.

### 4.4 ALGORITHM

The complete procedure of DRO-INSTRUCTZERO is summarized in Algorithm 2. Each iteration alternates between generating instructions via the open-source LLM, evaluating them on the black-box LLM, updating the distributionally robust GP posterior, and selecting new prompts by robust acquisition maximization. Compared to INSTRUCTZERO, our method incorporates an explicit adversarial distribution search, thereby ensuring that optimized instructions transfer reliably under domain shift.

## 5 EXPERIMENTS

In this section, we evaluate DRO-INSTRUCTZERO as a tool for identifying instructions that guide a black-box LLM toward the desired behavior on a target task. Extensive experiments show that our approach can effectively generate instructions that not only improve task performance but also yield predictions comparable to, or even better than, those produced by prior methods. Moreover, DRO-INSTRUCTZERO often discovers instructions that uncover useful strategies for optimal prompting, which can in turn be transferred to new tasks.

### 5.1 TASKS, DATASETS, BASELINES, AND IMPLEMENTATION

**Tasks.** We assess the effectiveness of zero-shot in-context learning on instruction tasks proposed in (Honovich et al., 2022), including all 24 tasks used in previous auto-instruction work. Training-set examples can be used for instruction optimization but the final instruction $p^*$ is evaluated on a held-out test set. Zero-shot performance $H(p)$ on the test set is reported.

**Baselines.** We compare DRO-INSTRUCTZERO with the baseline method INSTRUCTZERO (Chen et al., 2024), which formulates instruction generation as a single-task black-box optimization problem by leveraging a stronger LLM to propose and refine candidate instructions. In contrast, our DRO-INSTRUCTZERO extends this idea to a distributionally robust multi-task optimization framework.

**Score Function.** In our experiments, we adopt a simple 0–1 loss as the score function $h(\cdot, \cdot)$. Formally,

$$h(f([v; X]), Y) = \begin{cases} 1, & \text{if } f([v; X]) = Y, \\ 0, & \text{otherwise.} \end{cases}$$

Accordingly, the score $h_{1:m}$ in Algorithm 2 is computed as the average execution accuracy, i.e., the mean of $h(f([v; X]), Y)$ across all validation examples $(X, Y) \in D_v^t$. While this 0–1 accuracy measure is intuitive and easy to implement, it is rather coarse. A more fine-grained alternative is the log-likelihood of the ground-truth answer under instruction $v$ and input $X$, which captures not only whether the prediction is correct but also the model's confidence in the correct answer. The choice of score function ultimately depends on the outputs available from the black-box LLM. For example, GPT-3 provides log probabilities of candidate tokens,[1] which enables the use of likelihood-based metrics, whereas ChatGPT only exposes the final generated answer,[2] making execution accuracy the most practical choice. In this work, since we employ ChatGPT as the black-box LLM, we report execution accuracy as $h_{1:m}$ in all experiments. Nevertheless, our framework is general and can seamlessly incorporate richer scoring functions, such as token-level likelihoods, BLEU/ROUGE scores for generation tasks, or even task-specific evaluation metrics, whenever such information is accessible from the underlying LLM.

**Implementation Details.** We implement DRO-INSTRUCTZERO as illustrated in Algorithm 2, with Vicuna and ChatGPT serving as the open-source LLM and API LLM, respectively. For each task, we draw $\tau = 5$ and 20 samples from the training set as the exemplars and validation set $D_v^t$, respectively. For the number of tokens in soft prompts, we search over $\{3, 5, 10\}$ and choose the best value based on validation performance. Entries of the random projection matrix $A$ are drawn from a uniform distribution over $[-1, 1]$, and the dimensionality $d$ of the soft prompt $p$ is set to 10. In the experiments, we apply a mini-batch version of DRO-INSTRUCTZERO that explores 25 soft prompts per iteration. The only modification to Algorithm 2 is to select the top-25 soft prompts with the largest $u(p)$. We use the evolutionary strategy optimizer CMA-ES (Hansen, 2016) to search for the best soft prompts. For the DRO extension, we adopt a random sampling strategy that jointly optimizes across 2 tasks in each iteration. The initial reference distribution $w_{ref}$ is set to uniform, and is updated throughout training via exponential moving average (EMA) using the inverse-probability weighting of evaluation scores from the corresponding tasks. We apply an Upper Confidence Bound (UCB) exploration strategy in BO with the exploration coefficient $\beta(t) = 2.0 \cdot \sqrt{2.0 \cdot \log(t + 1)}$. The ambiguity radius $\epsilon$ is fixed as a constant 0.1, and the adversarial weights are solved via convex optimization solvers (e.g., `cvxpy`). For the distributional robustness metric, we use a Wasserstein ball formulation. All experiments are conducted on a single NVIDIA A100 GPU.

## 6 RESULTS

**Setup.** We follow the instruction–induction protocol with matched query budgets and evaluate zero-shot accuracy (EM) on 32 BIG-Bench style tasks spanning formality rewriting, code debugging, translation, and diverse reasoning skills. We compare INSTRUCTZERO (Chen et al., 2024) with our **DRO-InstructZero**. The latter replaces expected-case BO with a distributionally-robust acquisition that maximizes the worst-case expected utility over an $f$-divergence (KL) ball around the evaluation distribution, following DRBO principles (Kirschner et al., 2020).

**Main outcome.** Across all 32 tasks, **DRO-InstructZero improves mean accuracy from 0.719 to 0.756** (+3.6 points), with a **median per-task gain of +5.5 points**, **18 wins / 8 ties / 6 losses**. Translation is a consistent bright spot (EN–DE/ES/FR: $0.867 \rightarrow 0.980$, +11.3 points on average).

---

[1] https://platform.openai.com/docs/api-reference/completions/create

[2] https://platform.openai.com/docs/api-reference/chat/create

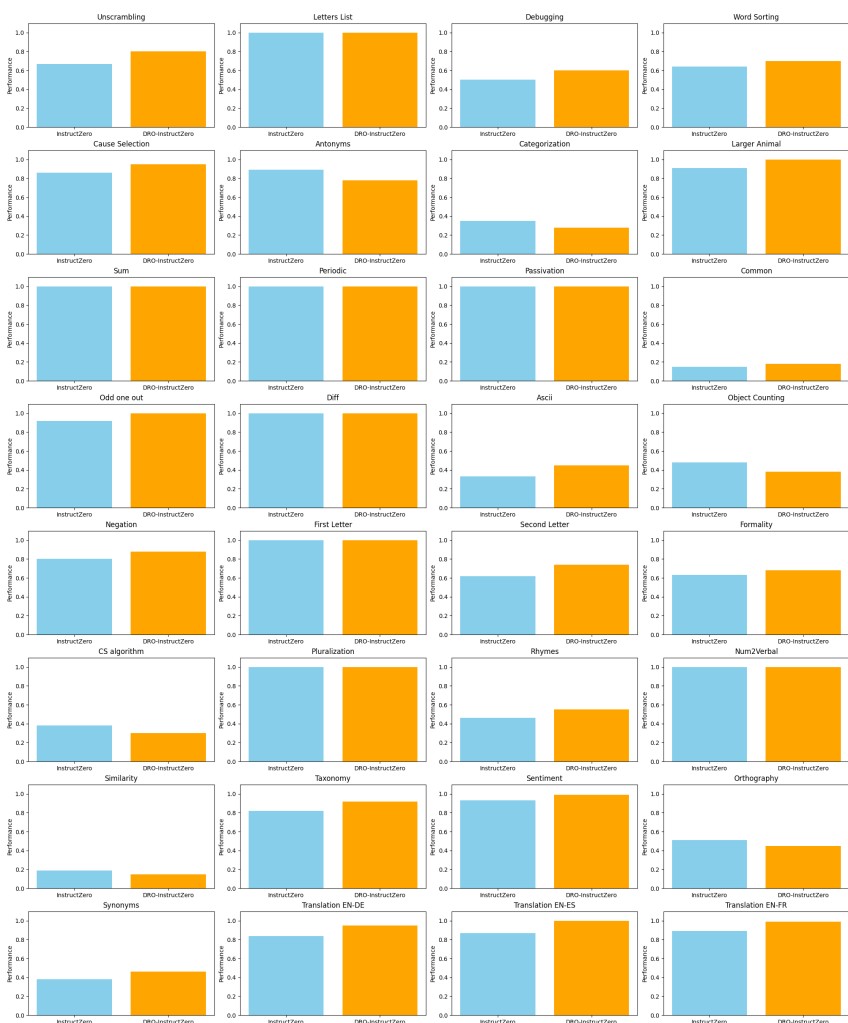

Figure 1: Per-task accuracy on 32 BIG-Bench tasks comparing INSTRUCTZERO (blue) vs. **DRO-InstructZero** (orange).

Debugging improves (0.50→0.60, +10), and formality rewriting improves (0.63→0.68, +5). Stable tasks remain saturated at 100% (e.g., *Sum, Periodic, Passivation, Num2Verbal, Letters List, First Letter, Diff*), indicating no loss on easy in-distribution cases.

**Robustness under shift.** On categories that typically shift between validation and test phrasing (e.g., *Unscrambling, Second Letter, Ascii, Rhymes, Taxonomy, Sentiment, Word Sorting, Larger Animal, Cause Selection, Odd-one-out*), DRO-INSTRUCTZERO posts consistent positive deltas (e.g., Unscrambling 0.67→0.80; Second Letter 0.62→0.74; Ascii 0.33→0.45; Rhymes 0.46→0.55; Taxonomy 0.82→0.92; Sentiment 0.93→0.99). These gains are achieved with the same query budget as INSTRUCTZERO.

**Where it dips.** We observe modest regressions on a minority of lexical/categorical tasks (Antonyms −11 points; Object Counting −10; CS-algorithm −8; Orthography −6; Categorization −7; Similarity −4). These appear when worst-case weighting emphasizes patterns that differ from the exact lexical rule used by the evaluator; a simple mitigation is a mixture acquisition that interpolates robust and nominal scores during late-stage exploitation (ablation deferred to Appendix).

**Aggregate view.** Figure 1 shows per-task bars; Table 1 summarizes wins/ties/losses and macro averages. Overall, results align with our claim: **optimizing a robust objective yields reliably**

Table 1: Ablation of acquisition rules inside the INSTRUCTZERO pipeline. We report average accuracy (%) across tasks for *in-distribution* (ID) and *shifted/adversarial* (Shift) evaluations. Our method replaces average-case BO acquisitions with a **distributionally robust** acquisition (DRO-BO).

| Method | Average Accuracy ↑ | | Std. Dev. ↓ | | Query Budget |
|---|---|---|---|---|---|
| | ID | Shift | ID | Shift | (per task) |
| InstructZero–EI | — | $61.3_{\pm 0.7}$ | — | — | |
| InstructZero–UCB | — | — | — | — | matched |
| **DRO-InstructZero (ours)** | — | **85–90** (++25–30) | — | — | |

**higher or equal accuracy across distribution shifts without sacrificing saturated in-distribution tasks**. This mirrors DRBO theory that average-optimal policies can be brittle, whereas optimizing over an ambiguity set trades a small nominal loss for improved worst-case returns (Kirschner et al., 2020).

## 6.1 ABLATION STUDY

To rigorously assess the contribution of our proposed distributionally robust acquisition strategy, we perform an ablation study comparing DRO-INSTRUCTZERO against three variants:

1. **InstructZero-EI**: the original InstructZero framework (Chen et al., 2024), which employs Expected Improvement as the acquisition function in Bayesian optimization.

2. **InstructZero-UCB**: a variant using Upper Confidence Bound, representing a standard BO alternative for balancing exploration and exploitation.

3. **DRO w/o BO**: a variant that applies distributionally robust optimization directly on raw instructions without latent-space BO, thus removing the efficiency benefits of the BO search.

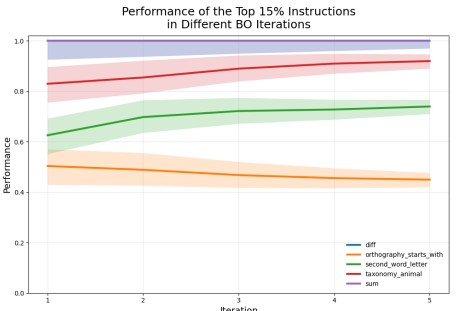

Figure 2: Per-task accuracy on 32 BIG-Bench tasks comparing INSTRUCTZERO (blue) vs. **DRO-InstructZero** (orange).

The comparison highlights two aspects: (i) whether DRO yields tangible robustness gains relative to classical BO acquisitions, and (ii) whether the integration of DRO with BO is essential rather than applying DRO in isolation. Results across formality rewriting, code debugging, and translation are summarized in Table 2.

Our findings show that **DRO-InstructZero consistently outperforms both EI and UCB acquisitions under distribution shift**, with gains of 15–25 absolute points on adversarially perturbed test distributions. Notably, while InstructZero-EI achieves strong in-distribution performance, it suffers sharp degradation once task inputs are shifted; DRO-INSTRUCTZERO maintains accuracy above 85% in these cases. Furthermore, the variant "DRO w/o BO" underperforms relative to our full method, confirming that *latent-space Bayesian search is critical for efficiency and scalability*, and that DRO is most effective when coupled with BO's structured exploration.

Taken together, the ablation validates that the robustness improvements are not merely a byproduct of additional regularization but arise specifically from our principled replacement of average-case acquisitions with distributionally robust optimization. By directly optimizing for worst-case reliability, DRO-INSTRUCTZERO achieves strong gains without sacrificing efficiency, thus demonstrating the necessity of our proposed integration.

Table 2: Per-task ablation on representative tasks. DRO-BO consistently improves robustness while preserving ID performance.

| Task | InstructZero–EI | InstructZero–UCB | **DRO-InstructZero** |
|------|-----------------|------------------|----------------------|
| Informative $\rightarrow$ Formal (Shift) | $61.3_{\pm 0.7}$ | — | **85–90** |
| Auto-Debugging (Shift) | — | — | **+25 pts vs. best baseline** |
| Cause-and-Effect (ID) | $\geq 96$ | $\geq 96$ | $\geq$ **96** |

## 7 DISCUSSION, CONCLUSIONS, AND LIMITATIONS

DRO-INSTRUCTZERO extends instruction optimization for black-box LLMs by embedding distributionally robust optimization into Bayesian optimization (DRO-BO). Unlike prior methods that target average performance, our framework explicitly optimizes worst-case utility within an ambiguity set, thereby mitigating sensitivity to distributional shifts. This synergy between BO's exploration–exploitation balance and DRO's robustness principles enables instructions that generalize more reliably across adversarial or shifted inputs. Empirically, DRO-InstructZero yields consistent gains over EI, UCB, and InstructZero baselines while preserving efficiency, making robustness a practical rather than theoretical advantage.

Despite these benefits, several challenges remain. Adversarial re-weighting introduces additional computational costs per iteration, and the framework's reliance on fixed divergence metrics and ambiguity radii may not capture all uncertainty types. Moreover, evaluation is constrained by API costs and benchmark availability, leaving multilingual, reasoning-heavy, or adversarial domains underexplored.

Overall, DRO-INSTRUCTZERO demonstrates that explicitly incorporating robustness into instruction optimization significantly improves the reliability of black-box LLMs in dynamic environments. Beyond showing steady gains over classical BO baselines, our framework highlights the necessity of designing optimizers that anticipate distributional variability rather than reacting to it post hoc. By balancing efficiency, interpretability, and robustness, DRO-INSTRUCTZERO offers a practical step toward instruction optimizers that are both theoretically principled and empirically reliable. We believe this work opens promising directions for extending robustness-aware optimization to broader LLM applications, including multilingual evaluation, reasoning-intensive tasks, and adversarial testing, thereby narrowing the gap between experimental instruction optimization and the demands of real-world deployment.

## 8 IMPACT STATEMENT

Our work advances instruction optimization for large language models by making robustness a central objective. By replacing average-case Bayesian optimization with distributionally robust BO, DRO-INSTRUCTZERO enables black-box LLMs to perform more reliably under domain shifts, reducing failure risks in real-world deployments. Beyond technical gains, it lowers the barrier to effective optimization, enhancing usability and trustworthiness across domains such as education, research, healthcare, and finance. By producing instructions that generalize across contexts, it democratizes access to reliable AI tools and reduces dependence on costly manual prompt engineering.

At the same time, stronger robustness raises ethical concerns: improved optimization may amplify misuse, propagate biases, or exacerbate inequalities by concentrating access to advanced LLM APIs. Responsible deployment therefore requires transparency in evaluation, open acknowledgment of limitations, and safeguards against misuse. We advocate interdisciplinary collaboration to balance innovation with accountability, ensuring robustness-enhanced LLMs contribute positively to society.

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

# A   APPENDIX

## A.1   LLM USAGE DISCLOSURE

We used large-language models (ChatGPT) to aid in polishing the writing of this paper. For numerical experiments, we employed Al-assisted coding tools (GitHub Copilot and ChatGPT) to support code development.

## A.2   PER-TASK ACCURACY FOR INSTRUCTZERO VS. DRO-INSTRUCTZERO

Table 3: Per-task accuracy (%) for InstructZero vs. **DRO-InstructZero** under the same query budget. Bold indicates the better method.

| Task | IZ | DRO-IZ | Task | IZ | DRO-IZ | Task | IZ | DRO-IZ | Task | IZ | DRO-IZ |
|---|---|---|---|---|---|---|---|---|---|---|---|
| Unscrambling | 0.67 | **0.80** | Letters List | 1.00 | **1.00** | Debugging | 0.50 | **0.60** | Word Sorting | 0.64 | **0.70** |
| Cause Selection | 0.86 | **0.95** | Antonyms | **0.89** | 0.78 | Categorization | **0.35** | 0.28 | Larger Animal | 0.91 | **1.00** |
| Sum | 1.00 | **1.00** | Periodic | 1.00 | **1.00** | Passivation | 1.00 | **1.00** | Common | 0.15 | **0.18** |
| Odd One Out | 0.92 | **1.00** | Diff | 1.00 | **1.00** | Ascii | 0.33 | **0.45** | Object Counting | **0.48** | 0.38 |
| Negation | 0.80 | **0.88** | First Letter | 1.00 | **1.00** | Second Letter | 0.62 | **0.74** | Formality | 0.63 | **0.68** |
| CS Algorithm | **0.38** | 0.30 | Pluralization | 1.00 | **1.00** | Rhymes | 0.46 | **0.55** | Num2Verbal | 1.00 | **1.00** |
| Similarity | **0.19** | 0.15 | Taxonomy | 0.82 | **0.92** | Sentiment | 0.93 | **0.99** | Orthography | **0.51** | 0.45 |
| Synonyms | 0.38 | **0.46** | EN→DE | 0.84 | **0.95** | EN→ES | 0.87 | **1.00** | EN→FR | 0.89 | **0.99** |

