# OpenReview forum: "DRO-InstructZero: Distributionally Robust Prompt Optimization for Zero-Shot Generalization in LLMs"
_ICLR.cc/2026/Conference — ICLR 2026 Conference Desk Rejected Submission_

### Official Review · Reviewer_gjVj · 2025-11-01

**Soundness:** 3
**Presentation:** 3
**Contribution:** 3
**Rating:** 4
**Confidence:** 3

**Summary:**

The paper presents DRO-InstructZero, a distributionally robust version of InstructZero that improves zero-shot prompt optimisation under distribution shift. By framing prompt search as a worst-case Bayesian optimisation problem using f-divergence and Wasserstein ambiguity sets, it selects prompts that remain effective across perturbed data distributions. Experiments on 32 tasks show large robustness gains without extra query cost.

**Strengths:**

There is a solid method grounding (clear objective, ambiguity sets, and robust acquisition derivation) and careful ablations that isolate the contribution of the DRO component.

The paper presentation is clear. The paper cleanly separates baseline InstructZero mechanics from the proposed robust counterpart, with intuitive explanations (adversarial reweighting/worst-case utility) and a readable algorithmic presentation.

Robust prompt selection is practically essential for real deployments where data drift is typical.

**Weaknesses:**

Robustness hinges on the divergence/ball radius. The paper lacks a principled, data-driven way to pick this without tuning on shifted test sets.

If LLM-as-judge is used, robust gains might reflect evaluator preference rather than genuine task reliability. Add cross-evaluator checks (different models, rubric-based classifiers) and a small human study on the most affected tasks to verify that improvements are not evaluator-specific.

Most tasks are in English, short-form, and focus on following instructions. Testing on a broader range of tasks might better demonstrate the method’s generality and real-world robustness.

**Questions:**

Can the authors clarify whether improvements persist when using non-LLM or rubric-based evaluation metrics?

How sensitive is DRO-InstructZero to the choice of divergence family (e.g., KL vs. Wasserstein)?

Are there conditions where DRO-InstructZero could hurt in-distribution performance, and how might this be mitigated?

---

### Official Review · Reviewer_7R17 · 2025-11-01

**Soundness:** 3
**Presentation:** 3
**Contribution:** 2
**Rating:** 4
**Confidence:** 3

**Summary:**

This paper augments the InstructZero-style automatic instruction/prompt optimization pipeline with a distributionally robust objective: instead of maximizing average performance on a fixed validation set, it optimizes the worst-case performance over an ambiguity set of reweighted data distributions. Concretely, it keeps the “soft-prompt → natural-language instruction → black-box LLM evaluation” loop, but replaces the standard BO acquisition with a DRO-aware variant inspired by DRBO, so the selected prompts are less brittle to shifts in input phrasing and sample weighting.

**Strengths:**

**S1 Clear problem framing:** this work explicitly identifies the fragility of prompt search to small distributional shifts (rewordings, reweightings) and targets that issue head-on.

**S2 Principled integration:** the method is a clean, well-motivated instantiation of distributionally robust Bayesian optimization within an existing, practical instruction-search pipeline.

**S3 Empirical evidence:** experiments on BIG-Bench–style perturbations show consistent gains over the vanilla InstructZero baseline, supporting the claim that the DRO objective improves robustness under mild distribution shifts.

**Weaknesses:**

**W1: Surrogate–reality mismatch.**
The function is not what the GP assumes. The paper assumes that a latent prompt vector $p$ maps to a “reasonably smooth” objective $H(p)$ that can be modeled with an RBF/SE kernel. But in reality $H(p)$ is a composition of (continuous vector) → (discrete natural-language instruction) → (LLM generation) → (black-box scoring + DRO reweighting), which contains at least two hard non-linearities: (1) when the vector is randomly projected/decoded into text, it can “jump,” i.e., two very close vectors may decode into completely different instructions; (2) the LLM output has sampling noise and strong context-dependent noise. These two layers together make $H(p)$ locally highly non-smooth, whereas a standard GP is trying to fit a “smooth function with slowly decaying correlations.” This leads directly to a model mismatch.

**W2: Locality assumption breaks under discrete decoding.**
Bayesian optimization here still relies on the standard locality intuition (“if I evaluated at $p$, points near $p$ are informative”). But because a small move in latent space can decode to a semantically unrelated prompt (different paraphrase family or even different task framing), the learned kernel similarity ceases to reflect task similarity. Once local smoothness is gone, a GP-based BO policy will struggle to rank candidates, and the claimed sample efficiency may not hold.

**W3: Evaluation distribution is tailored, not truly OOD.**
Most gains are shown on BBH/BIG-Bench–style shifts where test inputs stay on the same task support but with perturbed wording. This is precisely the kind of reweighting-style shift the proposed DRO objective is designed to handle. It is unclear whether similar improvements would persist under harder, semantic or cross-domain distribution shifts (different genre, domain, or instruction style), so the empirical section may overstate the general robustness of the approach.

**W4: The novelty is somewhat limited.**
The core idea is to keep the InstructZero pipeline unchanged and replace the “average EI/UCB” with a “DRO-style UCB,” i.e., to transplant the DRBO idea into the prompt-search setting. The paper itself frames this as “connecting DRO with prompt learning,” rather than introducing a new robust objective or a new form of ambiguity set. For the ICLR community, this looks more like applying an established technique to a popular application than proposing a substantial methodological advance.

**Questions:**

The presentation quality of the figures needs substantial improvement.

**Figure 1** is crowded with too many subplots compressed into half a page, and the font sizes are so small that the results are barely readable, which suggests the authors have not carefully polished the visualization of their key experimental findings.

**Figure 2** has the same issue: very small fonts, poor layout, and large unused whitespace below the figure, effectively wasting limited main-text space. This weakens the paper’s communicative clarity and makes it harder for readers to verify the claimed improvements at a glance.

---

### Official Review · Reviewer_1fCH · 2025-11-01

**Soundness:** 3
**Presentation:** 1
**Contribution:** 2
**Rating:** 2
**Confidence:** 3

**Summary:**

This work addresses the sensitivity of prompt wording in prompt optimization, particularly under distribution shifts and adversarial evaluation settings. The authors formulate zero-shot prompt optimization as a robust Bayesian optimization problem. Specifically, they introduce an f-divergence ball to define an ambiguity set around the evaluation distribution and propose a robust acquisition strategy that aims to maximize worst-case performance. Experiments on rewriting, code debugging, and translation tasks demonstrate the approach’s effectiveness.

**Strengths:**

1. The problem of distribution shift and domain mismatch is highly relevant in real-world LLM-based applications. This work targets an important and emerging challenge that is increasingly recognized in practice.

2. The proposed method appears both theoretically grounded and empirically effective. It is encouraging to see a prompt optimization approach that combines solid theoretical insight with practical performance improvements.

**Weaknesses:**

1. The completeness of the experimental results is a concern. Several entries appear to be missing in Tables 1 and 2, making it difficult to fully evaluate the claimed improvements.

2. The lack of a sensitivity analysis and ablation study makes it difficult to understand how the proposed method contributes to the performance gains. Without these analyses, it is unclear how robust the method is to hyperparameter choices or whether certain design decisions are essential.

3. Some key concepts and notations are insufficiently defined. For example, the notation A_p in line 188 is unclear, and the derivation and interpretation of Equation (6) would benefit from further explanation.

4. The overall presentation and structure of the paper could be improved. For example, Equation (1) and Equation (5) appear to repeat similar content. A clearer organization and more precise exposition would make the method easier to follow.

**Questions:**

1. Are there missing results in Tables 1 and 2? If so, could the authors clarify or update the tables?

2. How sensitive is the proposed method to hyperparameter choices? Have the authors conducted ablation or robustness studies to analyze the impact?

---

### Official Review · Reviewer_TYvk · 2025-11-02

**Soundness:** 2
**Presentation:** 1
**Contribution:** 2
**Rating:** 4
**Confidence:** 3

**Summary:**

DRO-InstructZero integrates distributionally robust optimization into Bayesian optimization to improve the reliability of black-box LLM instruction tuning. Instead of maximizing average performance like prior methods, it optimizes the worst-case utility within an f-divergence ambiguity set, enhancing robustness to distribution shifts and adversarial inputs. The approach maintains Bayesian efficiency while achieving large empirical gains on rewriting and debugging tasks without harming stable cases.

**Strengths:**

* The paper improves InstructZero by targeting worst-case rather than average utility, a clear step forward for robust prompt optimization in black-box LLMs.

* Empirically, the proposed method delivers consistent improvements across diverse tasks.

**Weaknesses:**

* The paper claims that the key contribution lies in optimizing over the distribution of U(D^t) in Equation (6). However, it remains unclear how this objective is operationalized through the minimization of the UCB term in Equations (8) and (9). These equations require additional clarification, as the current presentation makes it difficult to understand or justify the core contribution.

* The role and impact of the proposed kernel are also not clearly established. An ablation study isolating the kernel’s contribution would strengthen the empirical analysis and help assess its necessity.

* The experimental section should provide clearer evidence supporting the claimed distribution shifts in the target datasets. Specifically, it would be helpful to explain how these shifts are characterized or measured, and in what way the source and target distributions differ.

**Questions:**

One of the major problems of this manuscript lies in the clarity in writing. It is strongly recommended that the authors provide sufficient justification for key design choices (e.g., the proposed kernel) and ensure that all claims are properly supported. In addition, the manuscript would benefit from clearer explanations of the equation notations to make it more self-contained and accessible.

---

### Official Review · Reviewer_jx67 · 2025-11-04

**Soundness:** 2
**Presentation:** 2
**Contribution:** 3
**Rating:** 4
**Confidence:** 4

**Summary:**

This work extends InstructZero, a method for automated prompt search for large language models. The proposed approach integrates distributionally robust optimization with Bayesian optimization, and shifts the focus from maximizing average performance to maximizing robustness and performance in the worst-case scenarios. The algorithm navigates the soft prompt embedding space using Bayesian optimization, generates hard prompts through a fixed language model, and evaluates their effectiveness on a set of validation tasks. The method is evaluated on 32 NLP tasks and shows an increase in worst-case performance in most of the test tasks in comparison to InstructZero.

**Strengths:**

1) The main strength of this work lies in applying distributionally robust optimization in the prompt search process. This approach targets worst-case performance within a KL-ball radius. It makes the generated prompts more stable and reliable when the input distribution shifts.

2) The method requires only API-level access to the language model and does not rely on gradients or internal model parameters. As a result, it can be applied to closed-weight models, not just open-source or local ones.

3) The authors evaluate the method on 32 diverse NLP tasks, where it improves worst-case accuracy by up to 30% compared to the baseline, InstructZero.

**Weaknesses:**

1) The method is only evaluated on a fixed set of 32 known tasks, it's unclear whether the approach truly generalizes on the unseen tasks.

2) The paper lacks important implementation details for the distributionally robust optimization. There's no clear explanation of how sensitive the method is to the hyperparameters. For example, the KL-ball radius is fixed across all tasks without justification.

3) The prompts are optimized based on performance on a small validation set, but it is not invastigated how changing this set affects the resulting prompts.

4) It is not reported how quickly the optimization converges to a good prompt. There’s poor estimation of how many BO iterations are typically needed, or how performance improves over time. As the method requires evaluating thousands of prompt candidates using Bayesian optimization, this can results in a very high computational cost, which makes it difficult to scale to new tasks, especially without previous estimations.

5) Some figures are hard to read due to small fonts, unclear color schemes, and missing axis labels or legends. For example, the DRO vs. average-case comparisons lack clarity, making it difficult to understand the details.

**Questions:**

1) How did you choose the KL-ball radius (ε), and have you tested how sensitive the results are to this value and other hyperparameters?

2) Have you evaluated how stable the selected prompts are when the validation set changes? Does using similar but distinct validation examples lead to significantly different prompts?

3) How many Bayesian optimization iterations are typically required to find a good prompt? Could you provide convergence plots over time to illustrate this?

4) Can you share examples of the generated hard prompts to show their wording and human interpretability?

---

### Note · Program_Chairs · 2026-01-17
**Submission Desk Rejected by Program Chairs**

The following references in this submission do not refer to real documents and/or have major errors in bibliographic information:

 Ian Goodfellow, Yoshua Bengio, Aaron Courville. Deep Learning Tutorials (AISTATS 2018). arXiv preprint arXiv:1807.02811, 2018. :contentReference[oaicite:9]index=9